# Modelling Healthcare Associated Infections with Hypergraphs

Vivek Anand
vivekanand@gatech.edu
Georgia Institute of Technology
Atlanta, Georgia, USA

B. Aditya Prakash
badityap@cc.gatech.edu
Georgia Institute of Technology
Atlanta, Georgia, USA

## ABSTRACT

Healthcare Associated Infections (HAIs) like MRSA are a major threat to our hospitals and public health systems that significantly affect lives and resources. Unlike many common transmissible diseases, HAIs spread not only via direct person to person contacts but also indirectly through infected surfaces. Consequently, many of the standard epidemiological models like SIS, SEIR etc. cannot be used here. As a result, recently, many 2 Mode models were developed where people and surfaces behave differently allowing the pathogen to spread both via people and surfaces. However, these 2-Mode models are linear and cannot model non-linear contagions and group interactions effectively. In this paper, we present an agent based 2-Mode Hypergraph based Model and show that it is more expressive than the graph based benchmark both theoretically and experimentally.

## KEYWORDS

Hypergraphs, Healthcare Associated Infections, Agent Based Models, MRSA

**ACM Reference Format:**
Vivek Anand and B. Aditya Prakash. 2022. Modelling Healthcare Associated Infections with Hypergraphs. In *epiDAMIK 2022: 5th epiDAMIK ACM SIGKDD International Workshop on Epidemiology meets Data Mining and Knowledge Discovery, August 15, 2022, Washington, DC, USA.* ACM, New York, NY, USA, 5 pages.

## 1 INTRODUCTION

Healthcare Associated Infections (HAIs), like Methicillin-resistant Staphylococcus aureus (MRSA) are a growing scourge to global healthcare systems increasing mortality and annually costing 28-45 billion USD [11]. Unlike infectious diseases like Covid-19 and Pneumonia, whose infection load is transmitted mainly through person to person contacts, HAI dynamics are far more complex and additional rely on indirect contact i.e. contact between infected surfaces and people [7] for transmission.

Most of the existing work on modelling HAI dynamics has been primarily on 2-Mode graph models. In these models, infections can spread both via people and locations with correspondingly different propagation mechanisms [7]. However, these models are a gross oversimplification of the natural phenomenon as they only consider pairwise interactions an not group interactions. As a result, these models are unable to model the low variance of individual case counts for many HAIs. To resolve this issue, we propose the

HETEROGENEOUS-HYPERGRAPH-SIS model which is a generalization of the graph based HETEROGENEOUS-GRAPH-SIS. We theoretically show HETEROGENEOUS-HYPERGRAPH-SIS is more expressive and more experimentally accurate than HETEROGENEOUS-GRAPH-SIS.

Our main contributions are:

- *Definition of procedures* HETEROGENEOUS-HYPERGRAPH-SIS - We formally introduce a novel Hypergraph based SIS model.
- *Equivalences between* HETEROGENEOUS-HYPERGRAPH-SIS *and* HETEROGENEOUS-GRAPH-SIS- We theoretically show when the two models are equivalent to one another and demonstrate how it is useful in practice.
- *Experimental Evaluation on Real Hospital Traces* - Using real hospital contact data and synthetic case counts we show that the HETEROGENEOUS-HYPERGRAPH-SIS is able to better model the ground truth than HETEROGENEOUS-GRAPH-SIS.

## 2 RELATED WORK

### 2.1 HAI modelling

Modelling HAIs can be mainly split in two categories, Ordinary Differential Equation (ODE) model and agent based models.

ODE based models [13] and [14] are compartmental based models that deterministically allow for state transitions between the various modelled states of infection.

Agent based models [7] [9] [8] however, stochastically model the interactions between the various patients, locations and HCWs. [7] in particular proposed a 2-mode Graph model to incorporate the environmental mediation on the pathogen load. These models have been successfully used to evaluate factors underlying transmission [9] and in predicting case counts [8].

### 2.2 Spreading on Hypergraphs

Over the past few years, there has been increase work on spreading on hypergraphs. [3] examined how both community structure and nonlinear dependence on infection pressure affected the SIS spreading. [6] [10] analyzed the dynamics behind extinction thresholds and influence propagation over homogeneous hypergraphs. [2] combined both classic network based and hypergraph based contact pattern to demonstrate the importance of group interactions in influence propagation on real life location based social networks.

In our work, we seek to combine these two different spheres of work to use hypergraphs to model the group interactions and the nonlinear pathogens to better predict HAI incidence.

## 3 PRELIMINARIES

Both HETEROGENEOUS-GRAPH-SIS and HETEROGENEOUS-HYPERGRAPH-SIS are 2-mode models, i.e. there are two different types of nodes. Here, Locations and Healthcare Workers (HCWs) can transmit

*epiDAMIK 2022, Aug 15, 2022, Washington, DC, USA*

**Table 1: List of Notations**

| Variable | Description |
|---|---|
| $\alpha$ | Pathogen Shedding Rate |
| $\beta$ | Disease infectivity |
| $\delta$ | Recovery Probability |
| $\tau_{ijt}$ | Transfer ratio from node $j$ to node $i$ at time $t$ |
| $l_t$ | Pathogen load vector at time $t$ |
| $x_t$ | Infection State vector at time $t$ |
| $A_t$ | Adjacency Matrix at time $t$ |
| $R_t$ | Pathogen Transfer matrix at time $t$ |
| $g$ | Elementwise Nonlinear function |
| $n_{ijt}$ | Number of hyperedges that $i$ and $j$ appear at time $t$ |
| $P$ | Total number of patients |
| $H$ | Total number of Healthcare workers (HCWs) |
| $L$ | Total number of Locations |
| $N$ | Total number of agents ($P + H + L$) |

pathogen whereas patients can both transmit pathogen and get infected as well.

---

**Algorithm 1:** HETEROGENEOUS-GRAPH-SIS procedure

1 **Inputs:** $\Theta = \{\alpha, \beta, \delta, \{G_t = (V, E_t)\}_{t=1}^T, \{\tau_{ijt}\}_{i,j,t} \{A_t\}_{t=1}^T \}$

2 Initialize infection-states $x_1$, loads $l_1$

/* Compute pathogen transfer matrix */

3 Compute $R_t(i, j) = \begin{cases} \tau_{ijt} A_t(i, j) & \text{if } i \neq j \\ \tau_{ijt} & \text{if } i = j \end{cases}$

4 **for** $t = 1, ..., T$ **do**

    /* Add loads for each node */

5     Update loads $l_{t+1} = R_t l_t + \alpha x_t$

    /* Calculate next state for patients */

6     **for** *each patient $i$* **do**

7         **if** *$i$ is susceptible at time $t$ (i.e., $x_t(i) = 0$)* **then**

8             $i$ gets infected (i.e. $x_{t+1}(i) = 1$ with prob. min{1, $\beta l_t(i)$}

9         **else**

10             $i$ gets susceptible (i.e. $x_{t+1}(i) = 0$ with prob. $\delta$

---

### 3.1 HETEROGENEOUS-GRAPH-SIS

*3.1.1 Variable Description.* We model the spread of infections by a sequence of temporal graphs $G_1, G_2, ....G_T$ with $T$ number of timesteps. Graph $G_t$ models whether the $N$ nodes (Patients, HCW and Locations) interacted with each other at that timestep $t$. As a result, we can represent each $G_t$ as an $N \times N$ symmetric binary adjacency matrix $\mathbf{A}_t$ where $\mathbf{A}_t(i, j) = 1$ if $i$ and $j$ interacted at time $t$ and $\mathbf{A}_t(i, j) = 0$ otherwise. To model how the pathogen is transferred between various agents, we define $\tau_{ijt}$ as the transfer ratio of pathogen load from $j$ to $i$ at time $t$.

*3.1.2 Procedure Description.* In this procedure, the pathogen transfer matrices at each time step are calculated based on the various transfer ratios and their corresponding adjacency matrices. Then at each timestep, we iteratively accumulate pathogen load on each node and stochastically let the patients undergo state transitions

---

(Susceptible to Infected or Infected to Susceptible). This can be clearly seen in Algorithm 1.

---

**Algorithm 2:** HETEROGENEOUS-HYPERGRAPH-SIS procedure

1 **Inputs:**

    $\Theta = \{\alpha, \beta, \delta, \{G_t = (V, H_t)\}_{t=1}^T, \{\tau_{ijth}\}_{i,j,t,h}, \{A_t^h\}_{t=1}^T, g(.) \}$

2 Initialize infection-states $x_1$, loads $l_1$

    /* Compute hypergraph pathogen transfer matrix */

3 **for** *each hyperedge $h$* **do**

4     Compute $R_t^h(i, j) = \begin{cases} \tau_{ijth} A_t^h(i, j) & \text{if } i \neq j \\ 0 & \text{if } i = j \end{cases}$

5 Compute $R_t^{self}(i, i) = \tau_{iith} \;\; \forall i \in V$

6 **for** $t = 1, ..., T$ **do**

    /* Add loads from each hyperedge */

7     $l_{t+1} = \alpha x_t + \Sigma_{h \in H} g(R_t^h l_t) + R_t^{self} l_t$

    /* Calculate next state for patients */

8     **for** *each patient $i$* **do**

9         **if** *$i$ is susceptible at time $t$ (i.e., $x_t(i) = 0$)* **then**

10             $i$ gets infected (i.e. $x_{t+1}(i) = 1$ with prob. min{1, $\beta l_t(i)$}

11         **else**

12             $i$ gets susceptible (i.e. $x_{t+1}(i) = 0$ with prob. $\delta$

---

### 3.2 Why Hypergraphs?

Inside a hospital, nodes don't have only a pairwise relationship with each other (People can interact with each other in groups at a location). For other infectious diseases, understanding the various group settings is critical in modelling how a pathogen spreads [5] and it is equally critical here. Moreover, as HAIs are environmentally mediated, it is necessary to not only consider the people involved in the transmission (Patients and HCWs) but also the locations (surfaces) that can accumulate and transfer pathogen. As a graph only explicitly models pairwise relationships, these group interactions can be lost in the corresponding graph representation.

To model the group interactions, we use a hypergraph representation of the data. Each hyperedge within the hypergraph is in effect a photograph or a snapshot of a particular location at a particular period of time. In each hyperedge, both patient and HCW nodes can be present in addition to a single location node. This is to model the fact that at a given point in time, patients, HCWs and the location they are present at can accumulate and shed pathogen.

More formally, we can define a hyperedge $h \in H$ as $h \subset \{P \cup H\}$ & $l \mid l \in L$.

One example to show how this group information can be lost can be seen in Figure 1. If we focus on node $v_2$, we can clearly see from the hypergraph snapshot that $v_2$ has been with ($v_3$) and ($v_1, v_2$) at $L_1$ at different times. However, from the graph representation, we cannot determine if $v_2$ interacted with both $v_1$ and $v_3$ together or with each of them separately. Hypergraphs, unlike graphs, can unambiguously model this group information.

Note that if we use use a temporal graph with countable number of timesteps we could model the various group interactions as

well as the hypergraph model. However, there needs to be some resolution that needs to be chosen as the pathogen cannot spread instantaneously. A day in this context is a natural aggregation, has been used in works like [1], and therefore use it as our unit of time.

### 3.3 Heterogeneous-Hypergraph-SIS

*3.3.1 Variable Description.* We model the spread of infections by a sequence of temporal hypergraphs, $H_1, H_2, ...H_T$ with $T$ timesteps. Hypergraph $H_t$ models whether the $N$ nodes (Patients, HCWs and Locations) interacted with each other at that timestep $t$. Each hyperedge $h_t^i$ within $H_t$ indicates whether the nodes present within that hyperedge interacted in a group setting at time $t$. Consequently, we can express each hyperedge as a subgraph or as a binary adjacency matrix $A_t^h$ where $A_t^h(i, j) = 1$ indicates that $i$ and $j$ interacted with each other in hyperedge $h$ at time $t$ and $A_t^h(i, j) = 0$ indicates otherwise. The notation used here is almost identical to Heterogeneous-Graph-SIS. The only change is that we add a superscript $h$ to indicate the corresponding variable for each hyperedge.

*3.3.2 Procedure Description.* Like in Heterogeneous-Graph-SIS, we define pathogen transfer matrices. However, as we are dealing with hypergraphs, we need to define it for each hyperedge. Subsequently, we again compute the load accumulated on each node like in Heterogeneous-Graph-SIS, but here we additionally discount the load accumulated across each hyperedge with the function $g$. This is to account for the nonlinear pathogen. This allows us to model the loss of the pathogen between various group interactions. After the new loads are computed, patients undergo state transitions stochastically.

## 4 THEORETICAL EQUIVALENCY BETWEEN THE TWO MODELS

Both the Heterogeneous-Graph-SIS and the Heterogeneous-Hypergraph-SIS are quite similar to each other. The following results when taken together clearly show that Heterogeneous-Hypergraph-SIS is far more expressive and powerful than Heterogeneous-Graph-SIS.

In both the following theorems, for the sake of simplicity we assume that $\tau_{ijth_1} = \tau_{ijth_2} \forall h_1, h_2 \in H_t$. This is because there would be far too many variables to calibrate over if we have different transfer ratios for each hyperedges. Moreover, in the proofs for these theorems, we inductively ensure that the load vector ($l_t$) remains the same across both models at each timestep.

Theorem 1 (Linear Equivalence). *If $g$ is linear, Heterogeneous-Graph-SIS and Heterogeneous-Hypergraph-SIS are equivalent if and only if $\forall \, i, j \in V, \tau_{ijth} = \frac{\tau_{ijt}}{n_{ijt}}$.*

In simpler words, the transfer ratio for the two nodes across hyperedge $h$ depends only on the graph transfer ratio and the number of hyperedges the two nodes appear in.

Theorem 2 (Nonlinear Equivalence). *If $g$ is nonlinear, then Heterogeneous-Graph-SIS and Heterogeneous-Hypergraph-SIS are equivalent if and only if $\forall i, j \in V \; \tau_{ijth} = g^{-1}(\frac{\tau_{ijt} l_t(j)}{n_{ijt}}) \, / \, l_t(j)$ and $g$ is invertible.*

In simpler words, the transfer ratio for the two nodes across hyperedge $h$ depends on the graph transfer ratio, $g^{-1}$, the pathogen load and the number of hyperedges the two nodes appear in.

*Proof Sketch:* At time $t$ assume that pathogen loads are equal for both the hypergraph model and graph model. Then, inductively solve for parameters ($\tau$ and $g$) needed to make pathogen loads at time $t + 1$ equal.

Theorem 1 shows that Heterogeneous-Hypergraph-SIS is equivalent to Heterogeneous-Graph-SIS when $g$ is linear. This shows that the hypergraph model can do whatever the graph model can do.

Theorem 2, shows a similar equivalence between the two models. However, assuming that the two models are equivalent here is false. This is due to the presence of the pathogen load ($l_t$) in the equivalence term. In both procedures, the pathogen load is not known beforehand when we run the simulation. This shows that Heterogeneous-Graph-SIS is NOT equivalent to Heterogeneous-Hypergraph-SIS when $g$ is nonlinear. The reason why the pathogen load appears is due to the nonlinear function $g$ which does not allow the $l_t(j)$ to be cancelled out. It has been shown that if spreading is nonlinear, hyperedges are more influential seeds than individual nodes [10]. Here however, nonlinearity ensures that Heterogeneous-Hypergraph-SIS is more powerful and expressive than Heterogeneous-Graph-SIS.

Now that we know that Heterogeneous-Hypergraph-SIS is more powerful, we need to evaluate it experimentally.

## 5 EXPERIMENTS

### 5.1 Experimental Setup

*5.1.1 Dataset.* The dataset that we use for the following experiments is the Lyon Hospital Dataset [12]. This dataset was collected from 29 patients and 46 HCWs in Lyon, France. For five days, each person's location was tracked using wearable RFID sensors and two agents were said to be in contact if they were in close proximity with each other for more than 20 seconds.

*5.1.2 Graph Construction.* We construct a graph for each day in question. For a given day, for each interaction between agents, we add an edge between the agent nodes. As we do not have any information about the locations where these interactions took place, we simply add a placeholder location node for each separate interaction.

*5.1.3 Hypergraph Construction.* We construct a hypergraph for each day in question. For a given day, if multiple agents interact with each other within the same timeframe, we assume that they have interacted with each other and create a hyperedge with all of the agents present and a separate dummy location node.

*5.1.4 Model Calibration.* We calibrate our model using the ABCpy RejectionABC backend [4]. As we only have five days of graphs, we repeat our graphs/hypergraphs for a total of 35 times in sequence. Moreover, we calibrate on weekly average active case counts i.e. the current active infections as the calibration is too brittle otherwise. The parameters that are calibrated for both procedures are $\alpha, \beta, \delta$ and all of the $\tau$s. For Heterogeneous-Hypergraph-SIS we additionally calibrate $g$ between $[0, 1]$. Like in Section 4, we assume

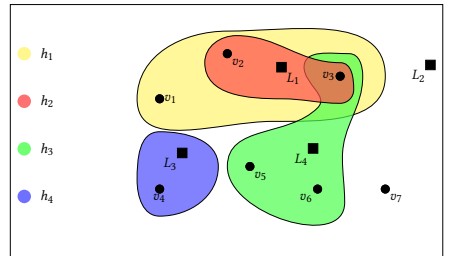

(a) Hypergraph Snapshot

| Node | Enter Time | Exit time | Location |
|------|-----------|-----------|----------|
| $v_3$ | 0 | 30 | $L_1$ |
| $v_1$ | 0 | 10 | $L_1$ |
| $v_2$ | 10 | 20 | $L_1$ |
| $v_5$ | 20 | 30 | $L_1$ |
| $v_6$ | 20 | 30 | $L_1$ |
| $v_4$ | 0 | 40 | $L_3$ |

(b) Sample Data

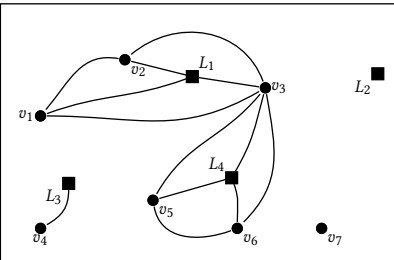

(c) Graph Snapshot

**Figure 1: Sample Trace Data and their corresponding Hypergraph and Graph representations**

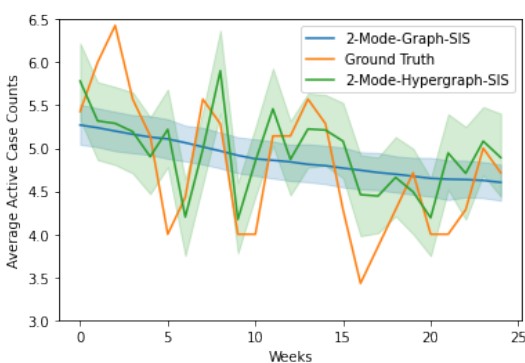

**(a) High Variance**

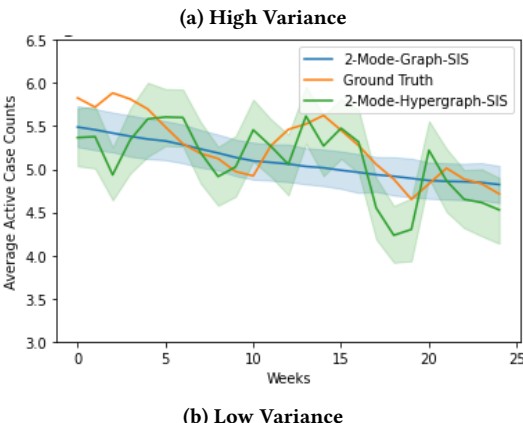

**(b) Low Variance**

**Figure 2: Plots comparing 100 runs of the best calibrated Heterogeneous-Graph-SIS, Heterogeneous-Hypergraph-SIS against the ground truth average active case counts**

a single set of $\tau$s for the entire simulation. We calibrate with a threshold $\epsilon = 1$.

*5.1.5 Aggregate Comparisons.* For this experiment, we will be comparing the relative performances of the graph and the hypergraph model on two different synthetic active case count datasets. The two datasets are described as High Variance and Low Variance respectively.

| Trace | 2-Mode-Graph-SIS | 2-Mode-Hypergraph-SIS |
|-------|-----------------|----------------------|
| High Variance | 0.436096 | 0.351075 |
| Low Variance | 0.062750 | 0.132840 |

**Table 2: Average RMSE for the best 2-Mode-Graph-SIS and 2-Mode-Hypergraph-SIS models against the ground truth average active cases**

*5.1.6 Synthetic Case Count Generation.* As we did not have access to actual case counts, we were forced to approximate real life HAI case counts. From conversations with other colleagues, we learned that real life weekly active cases could either be low or high variance. Consequently, we formulated such data to test our hypergraph model on.

## 5.2 Results

From Figure 2a) we can clearly see that Heterogeneous-Hypergraph-SIS models the High Variance ground truth significantly better than Heterogeneous-Graph-SIS. This is because Heterogeneous-Hypergraph-SIS has additional degrees of freedom to manipulate the threshold for $g$. This allows it to explore nonlinearity that Heterogeneous-Graph-SIS cannot. However, the variance is significantly more given that the 95% CI for Heterogeneous-Hypergraph-SIS is almost 1.5xs larger than Heterogeneous-Graph-SIS. This is to be expected given the greater number of parameters. From Table 2, we can see that not only is Heterogeneous-Hypergraph-SIS able to capture the various trends of the ground truth over time, but is also able to achieve a better RMSE.

From Figure 2b), we see that Heterogeneous-Hypergraph-SIS is somewhat able to model the trends in the ground truth whereas Heterogeneous-Graph-SIS fails to capture these trends. Again Heterogeneous-Hypergraph-SIS has more variance reflecting the same trend from Figure 2a). However, here, Heterogeneous-Hypergraph-SIS has a higher RMSE than Heterogeneous-Graph-SIS. This is likely due to the fact that the peaks in the ground truth are not large enough to impact the RMSE significantly enough.

## 6 CONCLUDING REMARKS AND FUTURE WORK

In this work, we have successfully been able to introduce a new hypergraph based model Heterogeneous-Hypergraph-SIS that has

been shown in our preliminary theoretical and experimental results to be more expressive than standard graph based models.

Though this work is promising, it is still in the preliminary stages and we still have to perform more experiments to comprehensively show that hypergraphs model HAIs better than graphs. In the future, we plan on expanding our scope of experiments to real life UVA hospital data.

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

## 7 ACKNOWLEDGEMENTS

This paper is based on work partially supported by the NSF (Expeditions CCF-1918770, CAREER IIS-2028586, RAPID IIS-2027862, Medium IIS-1955883, Medium IIS-2106961), CDC MInD program, faculty award from Facebook, and funds/computing resources from Georgia Tech. We'd additionally like to thank Jiaming Cui for useful discussions.