# OpenReview forum: "Modelling Healthcare Associated Infections with Hypergraphs"
_ACM.org/SIGKDD/2022/Workshop/epiDAMIK — KDD 2022 Workshop epiDAMIK Poster_

### Official Review · Reviewer_W5HT · 2022-06-21
**Early stages of an interesting work introducing hypergraphs as an alternative to model the group interactions  to better predict HAI**

**Rating:** 3
**Confidence:** 4

**Review:**

The paper introduces hypergraphs as an alternative to model the group interactions and the nonlinear pathogens to better predict Healthcare Associated Infections.
The paper can be considered as an acceptable presentation of early stages of a promising research agenda. There are, however, some instances that the authors make big claims without providing substantive support. An easy fix would be to modify the language a little bit. A more elegant addition for the more developed versions of the work would be providing theoretical guarantees and more comprehensive experimental support.
The following are some comments that I think might improve the paper.

Some comments:
- In Section 3.2 the authors claim that "To unambiguously model [...] group information, hypergraphs are necessary." I think "necessary" is a big word without well established theoretical results. And there might be other approaches that address the issue; so not that necessary.
- How are pathogen transfer matrices determined or identified?
-The authors assume similar transfer ratio from node 𝑗 to node 𝑖 at time t for different hyperedges claiming "This is a reasonable assumption to make as there will be far too many variables to calibrate over if we have different transfer ratios for each hyperedges." It is totally fine to make such assumptions but the provided argument should not be labeled as reasonable. If in reality these transfer ratios are very different and you assume they are the same for the sake of reducing complexity, it might not be reasonable. You can easily say for the sake of simplicity we assume .... .
-The authors state in the conclusions that "In this work, we have successfully been able to introduce a new hypergraph based model 2-Mode-Hypergraph-SIS that has been show both theoretically and in our preliminary experimental results to be more expressive than standard graph based models". I can see from Figure 2 (a) that hypergraphs outperform simple graphs. Also, two theorems are given which provide conditions for equivalence of hypergraphs and standard graphs. But I have difficulty finding where the paper "successfully" "theoretically" shows superiority of hypergraphs.

Minor comments:

Intro:
 far more complex and additional rely: additional -----> additionally
is able to better model the ground truth than: better  ------> use a more specific criteria to claim the proposed model is superior

HAI modelling:
various patients, locations and HCWs ------> the first time state what HCW stands for (healthcare workers?)

PRELIMINARIES:
Here, Locations: Locations ------> locations
Table 1 : be consistent in using lowercase and uppercase letters

2-Mode-Graph-SIS:
𝑁𝑥𝑁 symmetric binary: 𝑁𝑥𝑁----> 𝑁x𝑁

2-Mode-Hypergraph-SIS:
we need to define it fo each hyperedge: fo -----> for
but here w additionally discount: w -----> we?
This it to account for the nonlinear pathogen: it ----> is

CONCLUDING REMARKS AND FUTURE WORK
that has been show both theoretically and in our preliminary: show----> shown

---

### Official Review · Reviewer_DGAc · 2022-06-25
**Paper 14 clearly guides the reader through their modeling process but leaves a small amount of exposition to be desired in the motivation for a hypergraph structure.**

**Rating:** 4
**Confidence:** 4

**Review:**

This paper develops an algorithm to model the spread of hospital acquired infections between individuals and locations using hypergraphs. They contrast it with previous methods which only use graphs. They claim this new method yields an advantage with the presence of non-linearity in the model. The hyperedges are able to encapsulate more information about group settings than pairwise edges, yielding a more accurate representation of spread.

__Quality__: This paper is very well written and expertly navigates the topic at hand.

__Significance__: These results are potentially significant if they scale for more general datasets that reflect realistic spread over a longer period of time.

__Clarity__: Overall, this paper is presented in a very clear and concise way. The intentions are laid out in a very organized manner.

__Originality__: Models group spread as opposed to only pairwise spread.

__Strong points__
 - The way both graph and hypergraph methods are presented makes the comparison between their procedures clear.
 - The explanation for the difference between the graph and hypergraph data structures is clear. Figure 1 is helpful for visualizing this difference.
 - The exposition of what the authors are showing in this paper is very deliberate. They clearly state what they are showing and break down the problem concisely.

__Weak points__
 - As the reviewer understands it, the advantage of hypergraphs is through their ability to convey more information about interactions than only the temporal graph. Given this, are we able to simply expand the resolution of the graph procedure? Since the hyperedges each need an adjacency matrix representation, how much more complexity would be required to execute the graph algorithm on a higher number of timesteps?
 - A comparison of runtimes for each algorithm would be helpful in illustrating different settings each one may be appropriate.

__Minor__
 - Grammar: should be “increasing” instead of “increase” in first sentence, section 2.2. Rogue period at end of first column, section 4.
 - Spelling: “This allowis” at very beginning of page 4.

---

### Official Review · Reviewer_SgrR · 2022-06-27
**Review of Modelling Healthcare Associated Infections with Hypergraphs**

**Rating:** 1
**Confidence:** 4

**Review:**

The authors introduce a model of healthcare associated infections (HAIs), which spread not only person-to-person but also indirectly via infected surfaces. 2-mode models are typically used to capture how the pathogen spreads differently through people versus surfaces. In a 2-mode SIS graph model, agents (patients, healthcare workers, and locations) accumulate loads of the pathogen at each timestep through contact with infected agents and transition between susceptible and infected states accordingly. The authors introduce a 2-mode SIS hypergraph model, which captures group interactions, instead of only pairwise. The dynamics of the hypergraph model are highly similar to those of the graph model, except loads are accumulated through hyperedges (defined as group interactions between a subset of human agents and one location). The authors define when the graph and hypergraph models are equivalent and show that the hypergraph model is more expressive. They also calibrate both models on the Lyon Hospital Dataset, which captures interactions but is missing data on the location of interactions and actual case counts. They show that the hypergraph model can better fit synthetic aggregate case counts.

Strengths
+ Using hypergraphs to model HAIs seems new (based on authors’ literature review; I do not know all the literature in this area)
+ Promising results on synthetic data – hypergraph model better captures aggregate case counts

Weaknesses
- Their model is not particularly novel – SIS hypergraph models already exist (eg, Bodo et al, 2016) and, here, their HAI hypergraph model directly extends the dynamics of the HAI graph model
- Theorems are extremely straightforward and proofs are missing
- Experiments are limited – 1) location data is missing, despite the emphasis placed on 2-mode transmission. They create a new dummy location node for each interaction, which defeats the purpose of modeling transmission via locations. 2) Only synthetic aggregate case counts are available – given the high number of parameters, especially in the hypergraph model, are parameters identifiable with such coarse data? Also, where does synthetic data come from?

Minor suggestions
- Clarify introduction - the contributions seem to suggest that 2-mode graph models of HAIs are the authors' contribution (under contributions, "Definition of procedures 2-Mode-Hypergraph-SIS and 2-Mode-Graph-SIS- We formally define the two baseline procedures and their respective preliminaries") but then they explain and cite prior work that introduced 2-mode graph models of HAIs already (eg, Jang et al, 2019)
- HCWs - health care workers - are not defined when first mentioned (page 1)
- Typo: v2 should be with (v1, v3), not (v1, v2) on page 2
- Lots of notation used in theorems. Would be helpful to at least refer to the terms with a brief description when discussing theorems in text.

---

### Official Review · Reviewer_q3tC · 2022-06-27
**Interesting method, needs more exploration to prove efficiency.**

**Rating:** 2
**Confidence:** 4

**Review:**

This work aimed at demonstrating the 2-Mode Hypergraphs can model complex interactions more closely than the conventional 2-Mode Graph models. The concepts at hand are presented quite clearly and the purpose of hypergraphs to model all interactions more reliably is appropriately explained and represented in Figure 1. However, there are a few typos that should be corrected. The model is quite simple but could have interesting applications.
1. More information on the $g$ function could be provided. How does it enable modelling non-linear pathogens? Additional information could be presented regarding the calibration of $g$.
2. Generally, more information about the calibration process could have emphasized the novelty of the proposed technique.
3. The assumption that transfer ratios are constant across hyperedges is justified by the claim that there would be too many variables to calibrate otherwise. It would have been more relevant to add a practical reason. For instance, it might be assumed that thanks to rigorous protocols applied in hospitals, transfer ratios are likely constant across most locations and interactions.
4. The procedure to obtain a time-series of active cases count is also quite unclear. As noted, it would be a big plus to run a test of both models on actual data, allowing us to have more confidence on the efficacy of the hypergraph method. In fact, if the actual evolution of active cases count is closer to the “Low Variance” case, then the proposed method might not be the most relevant, considering the results of Table 2.
5. Please provide a GitHub repository of the code used for the experiments to allow replication of the analysis.